# Attention and social communication skills of very preterm infants after training attention control: Bayesian analyses of a feasibility study

**Oliver Perra**[1]*, **Fiona Alderdice**[1,2], **David Sweet**[3], **Alison McNulty**[4], **Matthew Johnston**[5], **Delfina Bilello**[6], **Kostas Papageorgiou**[5], **Sam Wass**[7]

**1** School of Nursing & Midwifery, Queen's University Belfast, MBC, Belfast, Northern Ireland, United Kingdom, **2** Nuffield Department of Population Health, University of Oxford, Oxford, United Kingdom, **3** Health and Social Care Belfast Trust, A Floor, Belfast City Hospital, Belfast, Northern Ireland, United Kingdom, **4** TinyLife, The Premature Baby Charity for Northern Ireland, Belfast, Northern Ireland, United Kingdom, **5** School of Psychology, Queen's University Belfast, Belfast, Northern Ireland, United Kingdom, **6** Institute of Mental Health, University of Birmingham, Birmingham, United Kingdom, **7** School of Psychology, University of East London, London, United Kingdom

* o.perra@qub.ac.uk

**Data Availability Statement:** Data cannot be shared publicly because of concerns about participants becoming identifiable. These

## Abstract

### Background

Very preterm (VP) infants (born 28 to <32 weeks of gestation) are at risk of cognitive delays and lower educational attainments. These risks are linked to anomalies in attention and information processing that emerge in the first years of life. Early interventions targeting attention functioning may equip VP infants with key building blocks for later attainments.

### Methods

We tested the feasibility of a randomised trial where VP infants took part in a computerised cognitive procedure to train attention control. Ten healthy VP infants aged approximately 12 months (corrected age) and randomly allocated with 1:1 ratio to the training (interactive computerised presentations) or an active control procedure completed the study. Before and after the training programme, participating infants completed a battery of screen-based attention tests, naturalistic attention and communication tasks, and temperament assessments. In a previous study we analysed the data concerning feasibility (e.g. recruitment and retention). In the paper presented here we considered the infants' performance and used Bayesian regression in order to provide credible treatment estimates considering the data collected.

### Results

Estimates indicate moderate treatment effects in visual memory: compared to controls, trained infants displayed improvements equivalent to 0.59 *SD* units. Trained infants also improved in their abilities to attend to less salient stimuli presentations by 0.82 *SD* units,

restrictions have been imposed by the the Health and Social Care Research Ethics Committee A (HSC REC A), Office for Research Ethics Committees Northern Ireland (ORECNI). Anonymised data will be made available from the https://pure.qub.ac.uk/en/datasets/ Institutional Data Access to qualified researchers. The non-author institutional point of contact to which data access queries may be sent must be Queen's University Belfast Governance, ethics and integrity, email: researchgovernance@qub.ac.uk.

**Funding:** Financial Disclosure Statement: this study was funded by a public health agency (PHA), health and social care (HSC) research and development division (https://research.hscni.net/) enabling research award to Dr O Perra (principal investigator) and co-investigators Prof F Alderdice, Dr S Wass, Dr K Papageorgiou, Mrs A Mcnulty, reference: STL/5274/16. The funders had no role in study design, data collection and analysis, decision to publish, or preparation of the manuscript.

**Competing interests:** The authors have declared that no competing interests exist.

compared to controls. However, results did not indicate relevant gains in attention habituation or disengagement. We also reported moderate improvements in focused attention during naturalistic tasks, and in directing other people's attention to shared objects.

## Discussion

The results warrant further investigation concerning the effectiveness of training attention control in VP infants, extending this line of research beyond our small and homogeneous sample of healthy VP infants. This study also emphasises the utility of Bayesian approaches in estimating potentially relevant effects in small samples or exploratory studies. The scope for further research on early attention control training is discussed in light of studies indicating VP children's susceptibility to positive environmental inputs.

## Trial registration

Registration ID: NCT03896490. Retrospectively registered at Clinical Trials Protocol Registration and Results System (clinicaltrials.gov).

## Introduction

Across developed countries 9.5 per 1,000 births are Very Preterm (VP), i.e. born between 28 and <32 gestation weeks [1]. Although survival and health prospects of VP children have improved since the 1990s [2, 3], they are still at increased risk for cognitive and learning difficulties [4–7], problem behaviour [8], and developmental disorders such as Attention Deficit with Hyperactivity Disorder [9–13].

Longitudinal studies of VP infants suggest that problems in learning and behaviour stem from difficulties in the way they regulate and control the acquisition of information from an early age [14–18]. A crucial role is played by VP infants' anomalies in attention control, the ability to voluntarily determine what to pay attention to, and what to ignore. This ability starts to emerge around the first year of life in typically-developing infants [19, 20], and marks a transition from infants' attention being highly reactive and directed externally by the environment, to their attention being directed voluntarily in accordance with the infants' goals [14]. Attention control is considered a key precursor of general top-down abilities to control cognitive processes in the service of goals such as Executive Functions (EFs) [21, 22]. Thus, early appearing anomalies in attention control may cause a cascade of consequences that affect the development of learning and self-regulation [18, 23–26].

Recent studies indicate that cognitive training interventions improve key EFs skills of children who were born VP [27–30]. Applying cognitive training interventions at an earlier age may produce larger transfer of effects [31], particularly if targeted at a key skill such as attention control. A similar intervention, the Attention Control Training (ACT), has been developed in typically developing infants [32–35]. The intervention involves showing interactive cartoons on a computer screen that move and change in response to infants' direction of gaze, thanks to an eye-tracking device that feeds information to the computer in real time. Infants engaged in the games receive "reward" presentations (e.g. display of characters making funny noises) if their visual behaviour meets increasing demands (e.g. maintaining attention on one character on the screen whilst ignoring an increasing number of visual distracters). Repeating

these tasks over time provides the opportunity to practise infants' attention control abilities in age-appropriate and engaging games.

We conducted a study to test the feasibility of delivering the ACT to VP infants [36]: in a previous study [37, 38] we presented the main results concerning feasibility (e.g. recruitment and retention): these results indicated that infants engaged in the training and assessments. Conversely, in this manuscript we investigate the direction of treatment effects across a series of infants' outcomes using Bayesian regression. Bayesian analyses allow to update assumptions about parameters (e.g. differences between two groups) by incorporating new information [39]. Bayesian approaches differ radically from conventional ones in the way they conceptualise parameters [40, 41]. In the conventional statistical approach, dubbed the *frequentist* approach, parameters are considered unknown but fixed: consequently, we cannot investigate the probability of a parameter of interest. Instead, we can only investigate the probability of a parameter being equal to the observed value (or more extreme ones) in a hypothetical counter-factual scenario that assumes a pre-defined value for the parameter, as in null-hypothesis testing. Furthermore, because the distributions of parameters are unknown, validity of inference in the conventional approach relies on the assumption that the sampling distribution of the parameter of interest approximates a known distribution (e.g. a normal distribution). This assumption is often plausible only when studies collect a large amount of information, e.g. by recruiting large numbers of participants.

In the Bayesian approach parameters are assumed to be uncertain, and thus can be described by probability distributions: these distributions can, in turn, be updated by incorporating newly observed data, thus providing more credible estimates. Inference in Bayesian analyses is thus informative and *valid* regardless of the sample size. Furthermore, by obtaining an updated probability distribution of parameters, it is possible to report statistics like *Uncertainty Intervals* (UI) that intuitively represent the range of values that the parameter of interest can credibly assume. Bayesian approaches are therefore well suited to provide valid and unambiguous inference even when few data are collected, and also provide straightforward methods for estimating the probability of replicating studies' results and power [42]: Bayesian approaches are thus apposite for the analyses of preliminary and exploratory studies or studies of small clinical samples. We started the analyses with a *sceptical* assumption of no differences in outcomes between treated and controls: these assumptions were updated considering the data we collected.

## Methods

This study was reviewed and approved by the Health and Social Care Research Ethics Committee A (HSC REC A), Office for Research Ethics Committees Northern Ireland (ORECNI), on 09 March 2018, REC Reference: 18/NI/0010; IRAS Project ID: 237537. The study has therefore been performed in accordance with the ethical standards laid down in the 1964 Declaration of Helsinki and its later amendments, as well as national laws concerning data protection. Caregivers of infants enrolled in the study provided written informed consent to take part in the study before initiating the study. The custodian of individuals pictured in Supplementary Material Section 3 (pages 6 and 7) in S1 File has provided written informed consent (as outlined in PLOS consent form) to publish their image alongside the manuscript.

We used Bayesian regression analyses to investigate the effects of the ACT programme in a small sample ($n$ = 10) of VP infants aged around 12 months from conception (in order to control for prematurity). Infants were randomly allocated to receiving the treatment or an active control procedure. Our overarching aim was to provide initial estimates of the intervention effects in order to inform future studies. Outcomes collected included visual attention and

visual memory assessed using screen-based computerised tasks. We also investigated focused and social attention, communication skills, and infants' temperament using naturalistic structured tasks to test transfer of effects in contexts and domains beyond those targeted by the training.

## Participants

Between 2018 and 2019 we recruited infants born very preterm who were under 12 months of age (corrected age) at time of recruiting. The key exclusion criteria were presence of significant cognitive or sensory and motor difficulties, or congenital anomalies (e.g. Cerebral Palsy). In Fig 1, we report a diagram detailing numbers assessed for eligibility, participants randomised and included in the analyses: Infants were allocated with 1:1 ratio to either the Attention Control Training (ACT) or a control procedure. Overall, 12 infants were recruited, but 10 (evenly split between treatment and control) completed the training/control procedure and the post-test assessments: One infant who had been randomised to the training dropped out before the scheduled pre-test; Another infant, randomised to the control procedure, dropped out after completing the pre-test. Participants' characteristics are reported in Table 1. We highlight that the main caregivers indicated higher than average educational attainments. Due to events beyond our control (e.g. families' commitments), infants in the control group were older at time of testing compared to treated ones. Thus, in data analyses we controlled for infant's age.

## Trial design

The study protocol and the feasibility outcomes have been described in previous publications [36–38]. Infants were allocated randomly to the ACT or a control procedure. Allocation was determined by computer-generated random numbers constrained to allow 1:1 allocation ratio in two blocks of 10. Allocation was provided in a sealed opaque envelope and was only known to the researcher responsible for the delivery of the training or control procedures after infants had completed the pre-tests. Pre- and Post-test assessments were conducted by a different researcher who remained "blind" to infants' group allocation. We also did not reveal group allocation to infants' parents. See [36–38] for more details.

Overall, infants completed five visits scheduled in consecutive weeks. The first visit involved a pre-test assessment followed by an attempt to deliver the first training or control session. In weeks 2, 3, and 4, the infant participated in the training or control procedure. In week 5, infants completed the same battery of tests as in the pre-test. The study received ethical approval by the relevant local health authorities. The study protocol was registered retrospectively in the Clinical Trials Protocol Registration and Results System (clinicaltrials.gov), registration ID: NCT03896490.

## Interventions

Infants in both groups sat on their parent's lap inside a photo-light tent, and watched cartoons displayed on a 19" screen. An eye-tracker (Tobii X-60) recorded infants' gaze direction and fed this information to a MacBook running Matlab.

**ACT.** The presentations to infants in the intervention group were interactive; The cartoon characters on the screen moved and changed in response to infants' visual behaviour in real time. Infants had to meet set criteria in order (e.g. follow a moving character across the screen) to receive a "reward" animation. These criteria changed adaptively, increasing demands when infants met lower-level criteria (e.g. a larger number of more salient distracters appeared on the screen when the infants displayed the ability to follow a target object across the screen). The training targeted three sets of skills: goal maintenance (e.g. maintaining attention on a

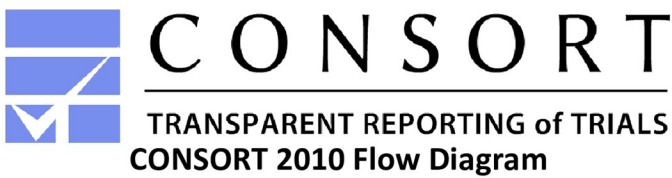

**CONSORT 2010 Flow Diagram**

Assessed for eligibility (n= 89 )

Meeting exclusion criteria, e.g. suspected Cerebral Palsy, (n=18 )

Eligible (n=71): Received study information and invitation to register interest in study with research group

No expression of interest (n= 37)

Registered interest (n=34)

Declined to take part (n= 22 )

Randomized (n=12)

Allocated to intervention (n= 6 )
- Received allocated intervention (n= 5  )
- Did not receive allocated intervention (Not attendance of pre-test and training: n= 1 )

Allocated to control  (n= 6  )
- Received allocated intervention (n=5  )
- Did not receive allocated intervention (Did not attend sessions after pre-test: n= 1  )

Lost to follow-up (Not attendance) (n= 1 )

Discontinued intervention (n= 0 )

Lost to follow-up (Not attendance: n= 1  )

Discontinued intervention (n= 0 )

Analysed  (n= 5 )
- Excluded from analysis  (n= 1 )

Analysed  (n= 5 )
- Excluded from analysis (n= 1  )

**Fig 1. CONSORT diagram.**

moving target); ability to identify a target among distracters; short term memory of objects embedded in scenes.

**Control.**   In contrast to the presentations for the intervention group, the presentations to infants in the control group were non-interactive: these followed a pre-set schedule, regardless

**Table 1. Participants' characteristics by group allocation (n = 10).**

|  | Controls | | Trained | |
|---|---|---|---|---|
| **Principal caregiver's employment status** | *n* | *%* | *n* | *%* |
| Employed Full-Time | 2 | *40* | 3 | *60* |
| Employed Part-Time | 2 | *40* | 2 | *40* |
| Not Employed | 1 | *20* | 0 | *—* |
| **Principal caregiver's educational attainment** | *n* | *%* | *n* | *%* |
| University Degree or Higher | 3 | *60* | 5 | *100* |
| **Child's sex** | *n* | *%* | *n* | *%* |
| Male | 4 | *80* | 3 | *60* |
| **Other characteristics** | **Mean** | *SD* | **Mean** | *SD* |
| Gestational Age (weeks) | 30 | *1.22* | 29.2 | *1.1* |
| Birth Weight (grams) | 1453 | *310* | 1313 | *338* |
| Days in Neonatal Intensive Care Unit | 42 | *14* | 57 | *21* |
| Age Pre-Test (Corrected) | 13.25 | *1.23* | 11.90 | *0.79* |
| Age Post-Test (Corrected) | 14.43 | *1.01* | 12.95 | *0.78* |
| Mullen Cognitive T score (Pre-Test) | 95.80 | *14.22* | 94.40 | *11.46* |

of infants' visual behaviour. In fact, each infant in the control group was matched to an infant in the control group and shown the pre-recorded presentation of stimuli produced by the matched infant in the intervention group. This ensured that presentations to infants in both groups were of similar length and type, except that the stimuli were not interactive for the control infants. Trained and control infants cumulative average duration of procedures completed were 84 min (*SD* = 9.11) and 75 min (*SD* = 8.60) respectively. More information on completion and engagement in the intervention and control tasks is provided in Supplementary Materials (SM)–Section 2 in S1 File, as well as a previous publication [37, 38]: analyses reported in the latter paper indicated that VP infants engaged with the training adequately.

## Pre- and post-test assessments

The assessments took place in a secluded room within the premises of a charity. Screen-based tasks were carried out using the same equipment and setting as the interventions. The other assessments took place around a desk and were video recorded by two cameras. An assessor blind to infants' group allocation delivered the tasks presented in four pseudo-randomised patterns, counterbalanced between and within participants. Details about the tasks, their coding and reliability, are provided in SM, Sections 3 and 4 in S1 File.

**Screen-based measures of attention.** *Sustained attention*. A habituation task where we recorded *peak looks* directed at a face (the duration of the single longest look to a stimulus) and *looks to criterion* (number of looks required to reach habituation).

*Visual recognition memory*. After habituation, a novel face was presented alongside the familiar one to calculate the *proportion of time looking at the novel target*.

*Gap-overlap*. We calculated *disengagement latencies* by comparing infants' reaction times to a simultaneous presentation of a lateral target alongside a central one vs. a presentation where the central target disappeared on appearance of the lateral one.

*Information density preference*. The outcome of interest was dubbed *attention capture* calculated as the difference between infants' look duration during a "fast", more salient presentation of stimuli, and a "slow" presentation. Increased attention for the less attention-eliciting slow presentations indicated infants' increased attention control.

**Attention during naturalistic tasks.**   We used the *Lab-TAB Orientation Task* [43, 44], and a *Semi-structured play task*, whereby infants played with wooden blocks and a set of toys respectively while being supervised by their parent. The tasks allowed to observe infants' focused attention in naturalistic settings. In the Lab-TAB the main outcome was infants' *intensity of facial interest*, rated by a blinded researcher. In the free play task, the outcome was the proportion of infants' looks directed at objects.

**Infant Behaviour Questionnaire (IBQ)–Very short form.**   The main caregiver completed the IBQ [45]: we focused on the "Effortful Control" dimension, which indicates infants' enjoyment of low-intensity activities (e.g. being held or rocking), focused attention, and inhibitory control.

**Social attention and communication.**   We administered the Object Spectacle, Book Presentation, and Gaze Following tasks from the Early Social Communication Scales (ESCS) [46]. We considered infants' proportions of trials where they displayed *Responding to Joint Attention* (RJA), i.e. they followed the experimenter's gaze and gestures directed at objects, and infants' rate of *Initiating Joint Attention* (IJA), i.e. instances where infants directed the experimenter's attention to objects.

**Attractive toy placed in a box task (Lab-TAB).**   We administered this task [43, 44] to measure infants' display of protest and anger, as rated by a blinded researcher. The researcher also coded parent's interference (e.g. attempts to soothe the infant): since this varied across groups, we weighted infants' display of anger by parental interference ratings (see SM–Section 4 in S1 File).

## Statistical analysis

We used Bayesian regression analysis [39] to estimate changes between pre- and post-assessments across treated and controls (An introduction to Bayesian approaches is provided in SM–Section 5 in S1 File). To this aim, we firstly calculated *difference scores* by subtracting individuals' scores in the pre-test from the post-test score: where necessary (e.g. Gap-Overlap task) these scores were inverse-transformed so that positive scores indicated improved performance after the training (see SM, Section 6 in S1 File). These were then standardised into *d* scores by considering the sample mean and *SD*: thus, $d = 0$ indicated no change between pre- and post-test, $d = 1$ indicated a 1 *SD* unit improvement in post-test performance, and so on.

For each outcome, we tested three models in sequence: the first model tested associations between treatment and difference scores, while the second model tested these associations while controlling for infants' age at post-test. A third model tested an interaction between treatment and infants' age, whereby we hypothesised that infants' age moderated the strength of the intervention effect. These models were compared using the Widely Applicable Information Criterion (WAIC) [47]: lower WAIC values indicate models with increased predictive value. We thus selected the model with the lowest WAIC, which we then used to estimate parameters of interest (see Table 5). Initial assumptions about the parameters were formalised in *prior* distributions: these assumed no differences between treated and controls (see SM–Section 7 in S1 File for details). The *posterior* distributions resulting from updating parameters distributions with data collected were simulated using Hamiltonian Monte Carlo (HMC) sampling implemented in the "ulam" command from the package "Rethinking" [39] in R [48]. We used four chains, each with a sample size of 20000 draws following a warm-up of 10000 draws.

Posterior distributions were used to estimate the average differences in *d* scores between treated and controls (the estimated distribution of these scores for treated and controls is displayed in SM–Section 8 in S1 File): we indicate them as $\hat{d}_T$. Thus, $\hat{d}_T$ scores represent the expected average post-test improvement of treated children compared to controls, expressed

**Table 2. Observed results of the screen-based tasks.**

| | | Sustained Attention | | | | | | Visual memory | | | Gap-Overlap | | | Information Density | | |
|---|---|---|---|---|---|---|---|---|---|---|---|---|---|---|---|---|
| | | Peak Look (in *seconds*) | | | Looks to Habituation | | | Proportions of looks to novel face (*Range 0 to 1.00*) | | | Difference latencies Overlap–Baseline (in *milliseconds*) | | | Diff. looking time slow vs. fast display (in *seconds*) | | |
| | | Pre-Test | Post-Test | *d* | Pre-Test | Post-Test | *d¹* | Pre-Test | Post-Test | *d* | Pre-Test | Post-Test | *d²* | Pre-Test | Post-Test | *d²* |
| Controls | Mean | 18.10 | 22.36 | +0.32 | 4.27 | 4.93 | -0.25 | 0.67 | 0.56 | -0.66 | 271.47 | 190.29 | +0.30 | 0.00 | 0.05 | -0.10 |
| | SD | *13.99* | *5.75* | *0.84* | *0.83* | *1.99* | *0.93* | *0.16* | *0.11* | *1.05* | *234.92* | *350.52* | *0.72* | *0.56* | *0.54* | *0.66* |
| Trained | Mean | 17.01 | 18.19 | +0.09 | 5.27 | 5.47 | -0.23 | 0.58 | 0.63 | +0.30 | 271.59 | 196.67 | +0.27 | 0.57 | 0.17 | +0.84 |
| | SD | *9.36* | *10.56* | *1.23* | *2.25* | *1.67* | *1.18* | *0.11* | *0.11* | *0.75* | *434.97* | *247.40* | *1.31* | *0.41* | *0.64* | *1.12* |
| Total | Mean | 17.55 | 20.28 | +0.21 | 4.77 | 5.20 | -0.24 | 0.62 | 0.59 | -0.18 | 271.53 | 193.48 | +0.28 | 0.29 | 0.11 | +0.37 |
| | SD | *11.24* | *8.31* | *1.00* | *1.68* | *1.74* | *1.00* | *0.14* | *0.11* | *1.00* | *329.57* | *286.04* | *1.00* | *0.55* | *0.56* | *1.00* |

[1] In accordance with previous studies (see SM, Section 3 in S1 File), in calculating the *d* score we used the reciprocal of the number of looks to habituation: Thus, *higher* scores in this outcome indicated *fewer* number of looks to reach criterion, hence *better* performance.

[2] In these outcomes lower scores reflected better performance (e.g. shorter latencies in turning to the target): to ensure all *d* scores followed the same pattern as other outcomes whereby *higher* scores indicated *better* performance, we report the inverse of these differences and used these inverse scores in further analyses.

in *SD* units: e.g. a positive $\hat{d}_T = 1$ estimated that the treated displayed a1 SD gain in post-test scores, on average, compared to the controls. When preferred models indicated an interaction between treatment and age, we estimated $\hat{d}_T$ scores at infants' age 13 months. We also report 89% Uncertainty Intervals (*UI*) which represent the range of $\hat{d}_T$ values expected with 89% probability. Finally, using the simulated posterior parameter distributions, we calculated the probability that $\hat{d}_T$ scores were > 0, i.e. the probability of average group differences favouring the treated.

## Results

We report descriptive results for the screen-based tasks in Table 2, and for the naturalistic tasks in Tables 3 and 4. Results concerning comparison and selection of models are presented in Table 5.

### Screen-based visual attention tasks

Results (see Table 2) indicated that the treated improved their performance in all these tasks, save for the looks to habituation outcome. Treatment-related changes were noticeable in

**Table 3. Observed results of naturalistic attention tasks.**

| | | Lab-Tab Orienting | | | Free Play | | | Infant Beh. Quest. | | |
|---|---|---|---|---|---|---|---|---|---|---|
| | | Intensity of Facial Interest. *Rated in a scale 0 to 2* | | | Proportion of time looking at objects (*0 to 1.00*) | | | Effortful Control: *Rated in a scale 1 to 5* | | |
| | | Pre-Tet | Post-Test | *d* | Pre-Test | Post-Test¹ | *d¹* | Pre-Test | Post-Test² | *d²* |
| Controls | Mean | 1.57 | 1.46 | -0.29 | 0.92 | 0.96 | +1.02 | 4.67 | 5.05 | +0.54 |
| | SD | *0.32* | *0.41* | *0.70* | *0.06* | *0.02* | *1.27* | *0.49* | *0.69* | *1.27* |
| Trained | Mean | 1.46 | 1.53 | +0.21 | 0.83 | 0.91 | +1.46 | 4.78 | 5.26 | +0.77 |
| | SD | *0.49* | *0.21* | *1.27* | *0.09* | *0.05* | *0.82* | *0.55* | *0.43* | *0.87* |
| Total | Mean | 1.51 | 1.49 | -0.04 | 0.88 | 0.93 | +1.26 | 4.73 | 5.17 | +0.66 |
| | SD | *0.39* | *0.31* | *1.00* | *0.09* | *0.05* | *1.00* | *0.49* | *0.53* | *1.00* |

[1] Note: Results are based on N = 9 because the post-test session of a control participant was not recorded due to malfunctioning equipment.

[2] Note: Results are based on N = 9 because the post-test questionnaire of a control participant was not completed.

**Table 4. Observed results of ESCS tasks and the Lab-Tab "Toy in a Box" task.**

| | | ESCS | | | ESCS | | | Lab-Tab *Toy in a Box* | | |
|---|---|---|---|---|---|---|---|---|---|---|
| | | Initiating Joint Attention: | | | Responding to Joint Attention: | | | Weighted average infants' display of protest: | | |
| | | Frequency of behaviour per minute | | | Proportion of trials with RJA (0 to 1.00) | | | Rated in a scale from 0 to 3 | | |
| | | Pre-Tet | Post-Test | *d* | Pre-Test | Post-Test | *d* | Pre-Test | Post-Test | *d* [1] |
| Controls | Mean | 0.056 | 0.058 | +0.07 | 0.49 | 0.50 | +0.06 | 0.46 | 0.42 | +0.12 |
| | SD | *0.042* | *0.039* | *1.35* | *0.14* | *0.13* | *0.97* | *0.29* | *0.21* | *1.03* |
| Trained | Mean | 0.053 | 0.061 | +0.24 | 0.43 | 0.41 | -0.07 | 0.33 | 0.54 | -0.59 |
| | SD | *0.016* | *0.033* | *0.63* | *0.14* | *0.19* | *1.14* | *0.18* | *0.44* | *0.93* |
| Total | Mean | 0.054 | 0.060 | +0.16 | 0.46 | 0.46 | -0.01 | 0.39 | 0.48 | -0.24 |
| | SD | *0.030* | *0.034* | *1.00* | *0.14* | *0.16* | *1.00* | *0.24* | *0.33* | *1.00* |

[1] Note: To ensure that higher *d* scores indicated better performance, we calculated the inverse of the difference between pre- and post-test. Thus, *positive* scores in *d* indicted ratings of infants' protest *decreased* at post- test.

Visual Memory and Information Density tasks: in contrast to treated infants, the controls displayed worsened performance between pre- and post-test. Model comparisons (see Table 5) indicated that age moderated the effect of treatment for peak looks, visual memory, and attention capture: differences between treated and controls diverged with increasing age, see Fig 2. We report estimates of *d* parameters and $\hat{d}_T$ scores in Table 6 and Fig 3. Overall, the results indicated negligible treatment effects in the Sustained Attention task and in the Gap-Overlap task ($\hat{d}_T = 0.02$ and $\hat{d}_T = -0.02$, respectively). Moderate effects favouring the treated were

**Table 5. Results of model comparisons using Bayesian regression.**

| Tasks→ | Sustained attention | | | | Visual Memory | | Gap-Overlap | | Info. Density | |
|---|---|---|---|---|---|---|---|---|---|---|
| Outcomes→ | Peak Look | | Looks to Habituation | | Novel Preference | | Disengagement | | Attention Capture | |
| Model | WAIC | SE | WAIC | SE | WAIC | SE | WAIC | SE | WAIC | SE |
| 1.Treatment | 32.7 | 4.57 | **32.5** | 3.56 | 30.0 | 3.86 | **32.3** | 2.68 | 30.6 | 6.11 |
| 2.Treatment + Age | 31.3 | 3.87 | 33.1 | 3.37 | 30.3 | 3.45 | 32.5 | 3.10 | 31.0 | 5.44 |
| 3.Treatment X Age | **30.8** | 3.74 | 33.4 | 3.31 | **28.1** | 3.60 | 32.9 | 3.03 | **30.0** | 4.79 |
| Tasks→ | Lab-TAB Orient. | | Free Play | | IBQ | | ESCS | | | |
| Outcomes→ | Facial Interest | | Looks at Objects | | Effortful Control | | Initiating JA | | Responding to JA | |
| Model | WAIC | SE | WAIC | SE | WAIC | SE | WAIC | SE | WAIC | SE |
| 1.Treatment | 32.3 | 5.26 | 29.7 | 2.25 | 29.9 | 3.76 | 32.9 | 5.07 | **32.4** | 3.29 |
| 2.Treatment + Age | 33.2 | 5.13 | 29.0 | 4.07 | **29.3** | 3.68 | 32.1 | 4.68 | 33.1 | 3.27 |
| 3.Treatment X Age | **30.3** | 5.26 | **28.9** | 3.54 | 29.5 | 3.73 | **31.4** | 4.65 | 33.2 | 3.15 |
| Tasks→ | Lab-TAB ToyBox | | | | | | | | | |
| Outcomes→ | Protest | | | | | | | | | |
| Model | WAIC | SE | | | | | | | | |
| 1.Treatment | **31.4** | 4.09 | | | | | | | | |
| 2.Treatment + Age | 31.6 | 3.82 | | | | | | | | |
| 3.Treatment X Age | 31.9 | 3.82 | | | | | | | | |

Note: For each outcome, we modelled: (1) A treatment effect model; (2) A model with independent treatment and age effects; (3) A model with an interaction between treatment and age, i.e. where treatment effects changed according to age at time of testing. These models were compared using the WAIC criterion (see Section 6 of Supplementary Material in S1 File) and we selected the model with lower WAIC, which indicated increased predictive value. Lowest WAIC values are highlighted in bold. SE = Standard Error.

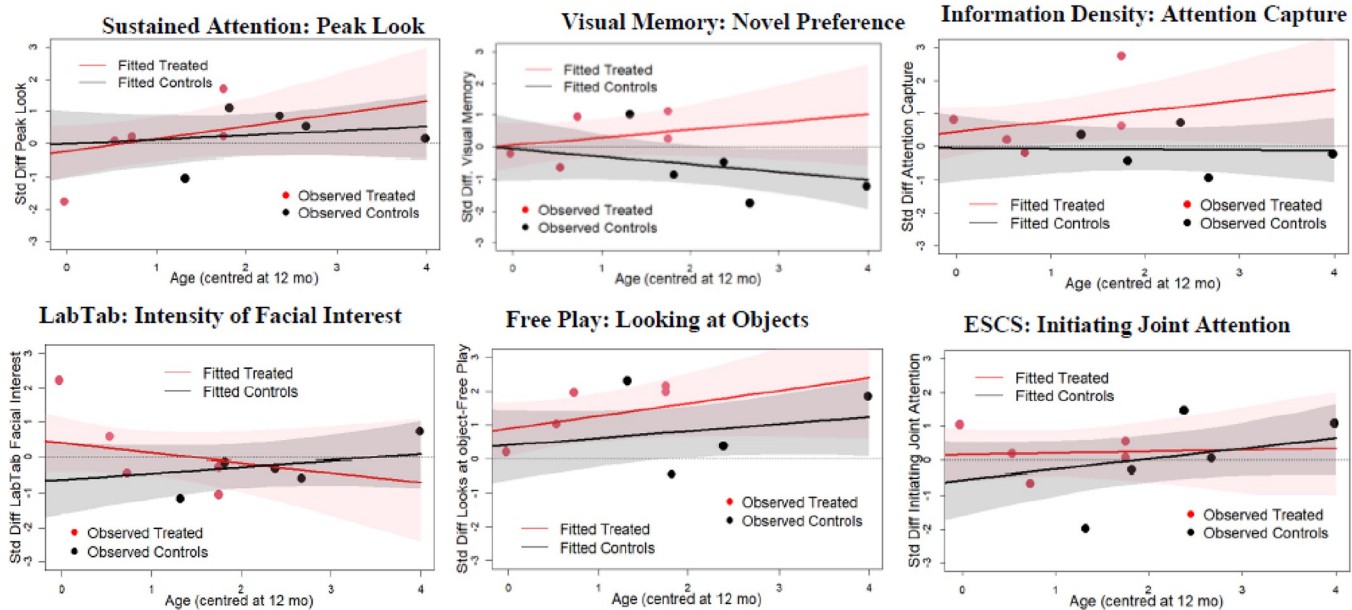

**Fig 2. Fitted (i.e.: Estimated) *d* scores by treatment and age in models that suggested relevant interactions treatment by age.** Lines represent estimated scores based on the posterior distribution. The shaded areas represent 89% uncertainty intervals of the scores estimated from the posterior distrition. Dots represent the observe treated and control scores by age.

observed in the Visual Memory ($\hat{d}_T$ = 0.59) and the Information Density task, ($\hat{d}_T$ = 0.82): although the *UIs* of these estimates were large, the probability of $\hat{d}_T$ scores in favour of the treated was 85% and 91% respectively.

## Naturalistic attention tasks

Results indicated that the trained infants displayed better performance compared to controls (see Table 3). Model comparisons indicated Age by Treatment effects for intensity of facial interest in the Lab-TAB orienting task and looks to objects during free play, see Table 5: Differences between treated and controls in facial interest diminished with age, while they increased with age when considering looks to objects, see Fig 2. Analyses indicated a moderate treatment gain ($\hat{d}_T$ = 0.57) in outcome intensity of facial interest, see Table 7 and Fig 4. A moderate difference in favour of the treated was also observed in infants' looks to objects during free play, $\hat{d}_T$ = 0.65. Although the *UIs* were large and imprecise, the probability of these differences favouring the treated were 82% and 84% respectively, see Fig 4.

## IBQ effortful control

Greater gains in this dimension were observed for treated infants, see Table 3. Analyses indicated a moderate difference in favour of the treated, $\hat{d}_T$ = 0.53, with the probability of these scores favouring the treated being 78%, see Table 7 and Fig 4.

## ESCS initiating and responding to Joint Attention (JA)

The treated displayed increased performance in Initiating JA, while the groups displayed similar scores in Responding to JA, see Table 5. Model comparisons suggested an interaction between age and treatment in Initiating JA whereby the effect of treatment did not vary across

**Table 6. Parameters estimated from the posterior distributions of models by screen-based outcomes.**

| Sustained Attention—Peak Look: | | | | |
|---|---|---|---|---|

$\hat{d}_T{}^1$ = 0.02 (89% UI -1.00 to 1.05). Probability of $\hat{d}_T$ > 0 estimated being 51%

| Parameter | d | SD | Lower 89% UI | Upper 89% UI |
|---|---|---|---|---|
| Intercept: Treated | -0.23 | 0.49 | -0.99 | 0.57 |
| Intercept: Controls | -0.01 | 0.63 | -1.02 | 1.01 |
| Rate of change: Treated | 0.38 | 0.31 | -0.12 | 0.87 |
| Rate of change: Controls | 0.13 | 0.24 | -0.25 | 0.52 |
| Sigma | 1.01 | 0.28 | 0.66 | 1.50 |

| Sustained Attention–(Reciprocal) Looks to Habituation: | | | | |
|---|---|---|---|---|

$\hat{d}_T$ = 0.02 (89% UI -0.98 to 1.01). Probability of $\hat{d}_T$ > 0 estimated being 52%

| Parameter | d | SD | Lower 89% UI | Upper 89% UI |
|---|---|---|---|---|
| Intercept: Treated | -0.18 | 0.45 | -0.89 | 0.53 |
| Intercept: Controls | -0.20 | 0.45 | -0.91 | 0.52 |
| Sigma | 1.12 | 0.28 | 0.76 | 1.63 |

| Visual Memory—Novel Preference: | | | | |
|---|---|---|---|---|

$\hat{d}_T$ = 0.59 (89% UI -0.33 to 1.52). Probability of $\hat{d}_T$ > 0 estimated being 85%

| Parameter | d | SD | Lower 89% UI | Upper 89% UI |
|---|---|---|---|---|
| Intercept: Treated | 0.06 | 0.44 | -0.63 | 0.77 |
| Intercept: Controls | -0.05 | 0.62 | -1.06 | 0.92 |
| Rate of change: Treated | 0.24 | 0.29 | -0.23 | 0.70 |
| Rate of change: Controls | -0.24 | 0.24 | -0.61 | 0.15 |
| Sigma | 0.89 | 0.25 | 0.58 | 1.34 |

| Gap-Overlap–(Inverse) Disengagement Latencies: | | | | |
|---|---|---|---|---|

$\hat{d}_T{}^1$ = - 0.02 (89% UI -1.02 to 0.98). Probability of $\hat{d}_T$ > 0 estimated being 49%

| Parameter | d | SD | Lower 89% UI | Upper 89% UI |
|---|---|---|---|---|
| Intercept: Treated | 0.22 | 0.45 | -0.50 | 0.92 |
| Intercept: Controls | 0.23 | 0.45 | -0.48 | 0.94 |
| Sigma | 1.12 | 0.29 | 0.76 | 1.64 |

| Information Density–(Inverse) Attention Capture: | | | | |
|---|---|---|---|---|

$\hat{d}_T{}^1$ = 0.82 (89% UI -0.15 to 1.77). Probability of $\hat{d}_T$ > 0 estimated being 91%

| Parameter | d | SD | Lower 89% UI | Upper 89% UI |
|---|---|---|---|---|
| Intercept: Treated | 0.45 | 0.46 | -0.29 | 1.18 |
| Intercept: Controls | -0.05 | 0.62 | -1.05 | 0.94 |
| Rate of change: Treated | 0.32 | 0.29 | -0.16 | 0.78 |
| Rate of change: Controls | -0.02 | 0.24 | -0.39 | 0.36 |
| Sigma | 0.96 | 0.27 | 0.63 | 1.44 |

*Note*: Different parameters included in the models for treatment and age were determined based on model selection results reported in Table 5. Where Treatment X Age interactions were included, these involved treatment-specific rate of change (also represented in Fig 2).

[1] The point-estimate difference was calculated at age 13 months (corrected age) in order to take into account age variation in the treatment effect.

UI = Uncertainty Interval. $\hat{\mathbf{d}}_T$ = Estimated average treated–control differences.

age, but controls displayed higher scores if older at time of testing, see Fig 4. We observed a moderate treatment effect in Initiating JA, $\hat{d}_T$ = 0.47, see Table 7 and a small difference favouring the controls in Responding to JA ($\hat{d}_T$ = - 0.11; 89% UI -1.12 to 0.89),. The probability of

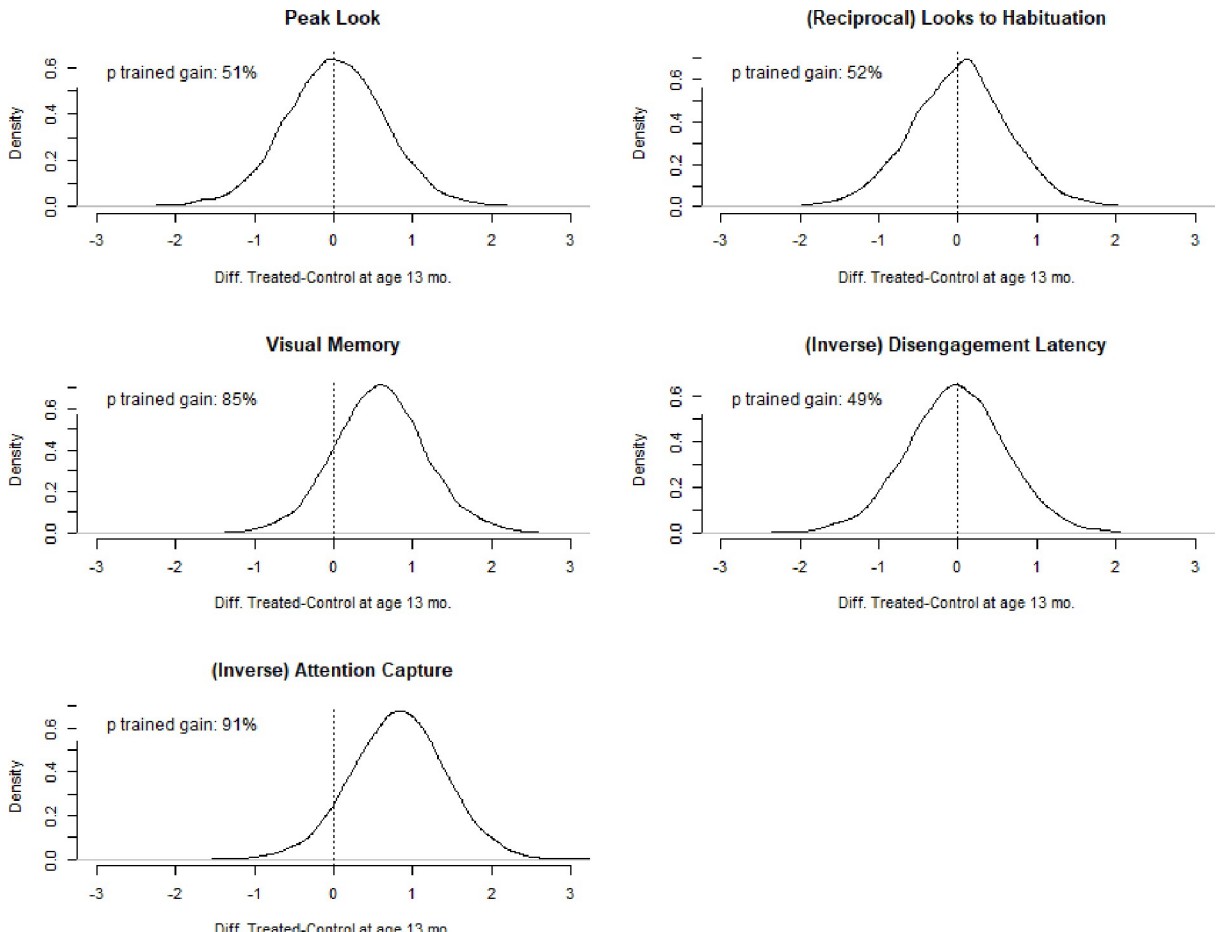

**Fig 3. Estimated distributions of $\hat{d}_T$ scores (i.e. differences in d scores between treated and controls) by screen-based outcomes.** The dotted lines highlight a $\hat{d}_T = 0$, i.e. no difference in performance between treated and controls. Scores on the right side of the dotted lines indicate improved post-test perfromance of the treated compared to the controls. NOTE: These scores were estimated by sampling n = 10,000 observations from the posterior distribution of parameters estimated in each selected model (see Table 5). Where the effect of the treatment was moderated by age, scores for treated and controls were estimated by considering scores at age 13 months.

these scores favouring the treated was 78% and 43% for Initiating and Responding to JA respectively.

## Intensity of protest

Results indicated an increase of protest displayed by the treated and a reduction for the controls, see Table 4. Analyses indicated a moderate effect of treatment that favoured the controls, $\hat{d}_T = -0.58$, see Table 7: The probability of the difference being positive was 16%.

## Discussion

The overarching aim of the study presented here was to provide initial estimates of VP infants' short-term changes across attention and socio-cognitive tasks after an early computerised cognitive training intervention. The intervention, delivered using a RCT design as part of a feasibility study, specifically targeted infants' attention control, which is considered a foundational skill for the development of top-down abilities in the service of learning and behaviour

**Table 7. Parameters estimated from the posterior distributions of models by naturalistic tasks outcomes.**

| Lab-TAB Orienting Task—Intensity of Facial Interest: | | | | |
|---|---|---|---|---|

$\hat{d}_T{}^1$ = 0.57 (89% *UI* -0.48 to 1.55). Probability of $\hat{d}_T > 0$ estimated being 82%

| Parameter | d | SD | Lower 89% UI | Upper 89% UI |
|---|---|---|---|---|
| Intercept: Treated | 0.41 | 0.48 | -0.40 | 1.14 |
| Intercept: Controls | -0.63 | 0.64 | -1.63 | 0.41 |
| Rate of change: Treated | -0.28 | 0.33 | -0.79 | 0.25 |
| Rate of change: Controls | 0.18 | 0.24 | -0.21 | 0.56 |
| Sigma | 0.96 | 0.29 | 0.59 | 1.49 |

| Free Play–Looks at Objects: | | | | |
|---|---|---|---|---|

$\hat{d}_T{}^1$ = 0.65 (89% *UI* -0.43 to 1.71). Probability of $\hat{d}_T > 0$ estimated being 84%

| Parameter | d | SD | Lower 89% UI | Upper 89% UI |
|---|---|---|---|---|
| Intercept: Treated | 0.89 | 0.49 | 0.10 | 1.66 |
| Intercept: Controls | 0.40 | 0.65 | -0.64 | 1.43 |
| Rate of change: Treated | 0.37 | 0.30 | -0.12 | 0.85 |
| Rate of change: Controls | 0.21 | 0.24 | -0.17 | 0.60 |
| Sigma | 1.06 | 0.31 | 0.68 | 1.61 |

| IBQ–Effortful Control: | | | | |
|---|---|---|---|---|

$\hat{d}_T$ = 0.53 (89% *UI* -0.63 to 1.63). Probability of $\hat{d}_T > 0$ estimated being 78%

| Parameter | d | SD | Lower 89% UI | Upper 89% UI |
|---|---|---|---|---|
| Intercept: Treated | 0.42 | 0.47 | -0.34 | 1.16 |
| Intercept: Controls | -0.11 | 0.68 | -1.18 | 0.99 |
| Rate of change | 0.25 | 0.23 | -0.13 | 0.61 |
| Sigma | 1.09 | 0.31 | 0.71 | 1.64 |

| ESCS- Initiating Joint Attention: | | | | |
|---|---|---|---|---|

$\hat{d}_T{}^1$ = 0.47 (89% *UI* -0.60 to 1.48). Probability of $\hat{d}_T > 0$ estimated being 43%

| Parameter | d | SD | Lower 89% UI | Upper 89% UI |
|---|---|---|---|---|
| Intercept: Treated | 0.17 | 0.49 | -0.61 | 0.93 |
| Intercept: Controls | -0.56 | 0.67 | -1.61 | 0.54 |
| Rate of change: Treated | 0.04 | 0.30 | -0.43 | 0.52 |
| Rate of change: Controls | 0.30 | 0.25 | -0.11 | 0.70 |
| Sigma | 1.03 | 0.28 | 0.67 | 1.54 |

| Lab-TAB Toy in Box–(Inverse) Infant's Protest: | | | | |
|---|---|---|---|---|

$\hat{d}_T$ = - 0.58 (89% *UI* -1.52 to 0.37). Probability of $\hat{d}_T > 0$ estimated being 16%

| Parameter | d | SD | Lower 89% UI | Upper 89% UI |
|---|---|---|---|---|
| Intercept: Treated | -0.49 | 0.43 | -1.15 | 0.22 |
| Intercept: Controls | 0.10 | 0.43 | -0.58 | 0.77 |
| Sigma | 1.05 | 0.27 | 0.71 | 1.53 |

*Note*: Different parameters included in the models for treatment and age were determined based on model selection results reported in Table 5. Where Treatment X Age interactions were included, these involved treatment-specific rate of change (also represented in Fig 2).

[1] The point-estimate difference was calculated at age 13 months (corrected age) in order to take into account age variation in the treatment effect.

UI = Uncertainty Interval. $\hat{\mathbf{d}}_T$ = Estimated average treated–control differences.

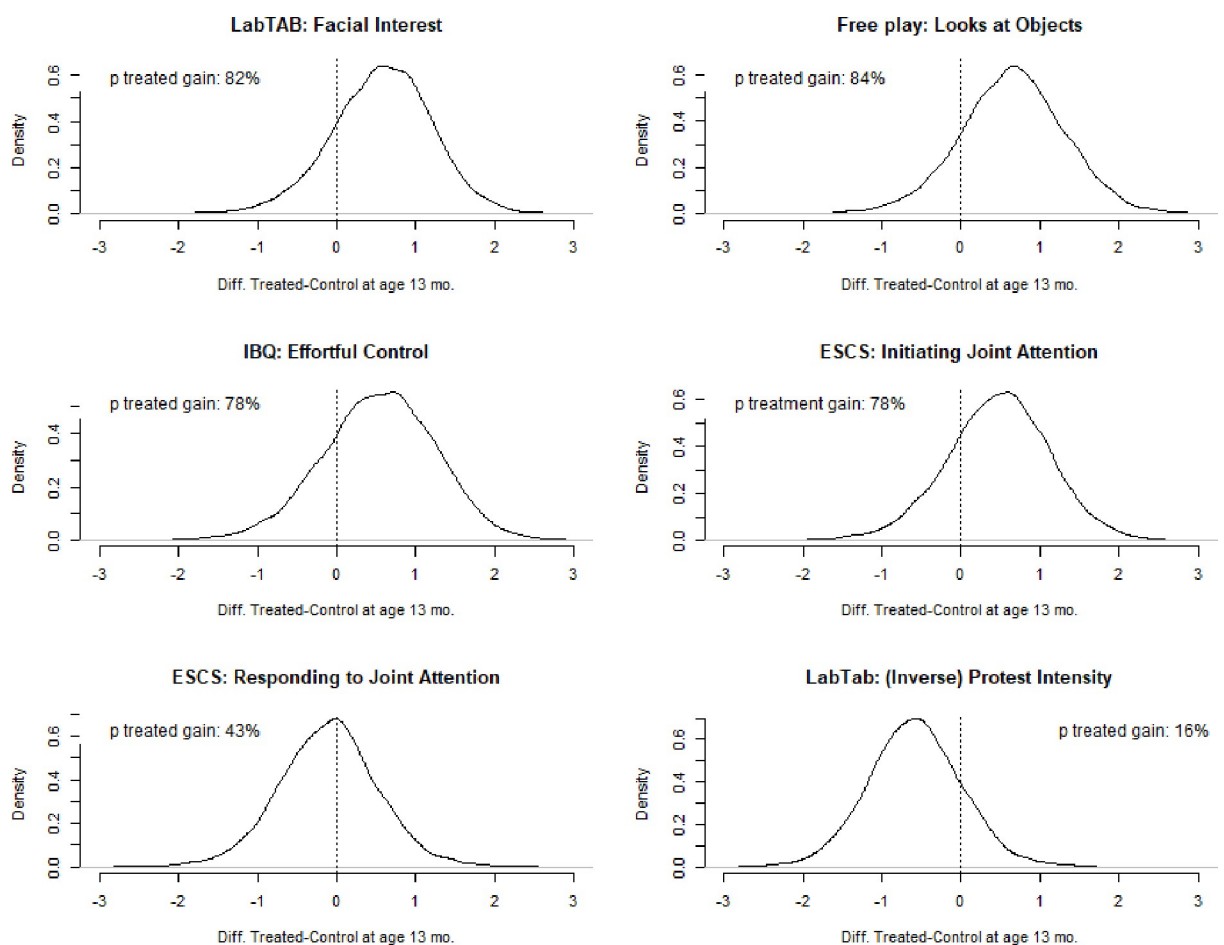

**Fig 4. Estimated distributions of $\hat{d}_T$ scores (i.e. differences in d scores between treated and controls) by naturalsitic tasks outcomes.** The dotted lines highlight a $\hat{d}_T = 0$, i.e. no difference in performance between treated and controls. Scores on the right side of the dotted lines indicate improved post-test perfromance of the treated compared to the controls. NOTE: These scores were estimated by sampling n = 10,000 observations from the posterior distribution of parameters estimated in each selected model (see Table 5). Where the effect of the treatment was moderated by age, scores for treated and controls were estimated by considering scores at age 13 months.

regulation [22, 49, 50]. VP infants were approximately 12 months (corrected age) when taking part in the study, an age when attention control abilities are considered to be emerging. We hypothesised the intervention could improve infants' performance in the short term, particularly in attention tasks more akin to the abilities trained (e.g. sustained attention). To provide a robust test of our hypotheses, in our analyses we initially assumed a *sceptical view*, postulating no differences between treated and controls: using Bayesian methods, these assumptions were updated considering the data we collected.

VP infants who received the training did not display noticeable gains in tasks such as habituation and disengagement. This contrasted with results from two previous studies of typically-developing infants who received the ACT programme for an equivalent length of time and number of sessions as in this study [31, 32]. In particular, while the effect of the ACT on sustained attention in our study was negligible ($\hat{d}_T = 0.02$, *89% UI* -1.00 to 1.05), Wass and colleagues [31] and Ballieux and colleagues [32] had reported moderate treatment effects in this task: *Cohen's d* = 0.65 and *Cohen's d* = 0.69, respectively. In our study we reported a negative but negligeable effect of the intervention on disengagement latencies in the Gap-Overlap task

($\hat{d}_T$ = - 0.02, *89% UI* -1.02 to 0.98), while previous studies reported a moderate positive (*Cohen's d* = 0.68, see [31]) and small positive effect (*Cohen's d* = 0.35, see [32]). Future studies comparing term and VP infants' performance in these tasks across time and different phases of the intervention are necessary to investigate attention differences and their underlying mechanisms across these groups.

Our study, however, did indicate moderate treatment effects in the ability to memorise complex visual stimuli ($\hat{d}_T$ = 0.59; 89% *UI* -0.33 to 1.52), and the ability to maintain attention on less salient and less automatically attention-eliciting stimuli presentations ($\hat{d}_T$ = 0.82; 89% UI -0.15 to 1.77). Moderate treatment effects were also observed in naturalistic tasks (e.g. focused attention during object play: $\hat{d}_T$ = 0.65; 89% UI -0.43 to 1.71) and in parental reports of infants' effortful control ($\hat{d}_T$ = 0.53; 89% UI -0.63 to 1.63). An aversive treatment effect ($\hat{d}_T$ = - 0.58; 89% UI -1.52 to 0.37) was reported insofar treated infants were more prone to protesting in a naturalistic frustrating situation: However, infants displayed low levels of protest during this task, as demonstrated by the average ratings of this behaviour being close to 0 (see Table 4).

This is the first study that trained VP infants' attention control. Previous analyses indicated it is possible to recruit and retain VP infants and engage them in the training [37, 38]. Conversely, the analyses of infants' performance in different tasks presented in this paper are important in providing initial estimates concerning the potential for improving early key cognitive abilities of this at-risk group. The treatment effect in visual memory was estimated to be as large as 0.59 *SD* units: this suggests treated infants displayed better abilities to maintain task-relevant information for brief periods of time. Performance in memory tasks in infancy can predict working memory in adolescence [51]: improvements in memory may thus be an important foundation for later attainments. A moderate treatment effect of 0.82 *SD* units was also estimated in infant's abilities to overrun a preference for 'automatically' attention-eliciting stimuli presentations in favour of less salient ones. This ability may indicate a shift from attention being exogenously controlled to attention being controlled endogenously (from the inside) and voluntarily. The results indicate that treatment effects were moderated by infants' age in the study (see Fig 2): improvements were larger with increasing age of the treated. Whilst early intervention may favour VP infants' development, these results suggest the need to consider the presence of *sensitive periods*: VP infants may be more receptive to attention control training with increased maturation.

Moderate treatment effects observed in naturalistic tasks are suggestive of potential for transfer of effects. Treated infants displayed improvement in *focused attention* during naturalistic object play. Parents of infants in the intervention also reported improvements in infants' *effortful control* in a validated questionnaire, a dimension that taps into infants' focused attention (e.g. prolonged play with objects), and inhibitory control (e.g. infants' soothing in response to parental talking). Parents were not told whether their child was receiving the treatment or the active control procedure, and the results we reported in a previous paper [37, 38] showed that parents did not accurately guess in which group their child had been allocated: we thus argue it is unlikely that parental ratings were biased by their awareness of group allocation. Overall, these findings suggest treated infants being able to focus attention during their interactions with the environment. A study with a similar population reported that preterm infants' focused attention predicted cognitive performance in childhood [17]: replication and extension of our results would thus suggest encouraging prospects for treated infants.

A moderate treatment effect was also observed in infants' *Initiating Joint Attention* (IJA), the ability to employ their direction of gaze and gestures to spontaneously share experiences with others. IJA has been linked to voluntary attention process underpinned by frontal systems

[52–54]. Joint attention, in turn, is pivotal in enabling infants' social learning [55]: further evidence of effect transfer into joint attention skills would thus strengthen the case for early training of attention control playing a *gating* role in the onset of further learning skills.

The results concerning treated infants' increased protest in a frustrating situation may induce caution. However, descriptive results indicate the protest behaviour in the task was of low intensity overall. Furthermore, other studies suggest that protest in frustrating situations is a normative reaction when infants develop abilities to explore the environment [56]: infants' protest might rather be interpreted as a sign of infants' increased engagement with the task.

## Limitations

The significance of the findings reported is limited by the small sample size in this study and its homogeneity: we recruited generally healthy VP infants; Furthermore, VP infants in the study came from families with higher than average educational attainments. While the results of Bayesian analyses are *valid* and *informative*, the estimated effects might not be replicated in other samples: the sample size of this study was very small and the estimated effects might not be generalisable. Furthermore, the estimated posterior parameters lacked precision: these displayed large uncertainty intervals (*UIs*) which, in all cases, included values that indicated no difference between groups' outcomes. Further caution is necessary considering that trained infants did not display improvements in key attention skills such as sustained attention [57] or disengagement [22, 58]. Finally, the outcomes were collected soon after the training or active control procedures and we did not follow up the participants afterwards.

## Conclusions

While the results of this study should not be over-estimated, the use of a Bayesian approach is useful in providing credible estimates of effects generated by combining the evidence collected with explicit (i.e. formalised) sceptical assumptions (see SM–Section 5 in S1 File). This approach is valuable in estimating potential effects in a small sample like ours: advances in computing systems, as well as the development of statistical packages and tutorials on the implementation of Bayesian approaches, make these more accessible for exploratory studies, studies of small clinical groups, and so on. While the traditional frequentist approach provides "impoverished" estimates of parameter values with ill-defined confidence intervals [42], Bayesian approaches provide unambiguous information about the relative credibility of parameter estimates as well as their trade-offs. The use of Bayesian approaches is also consistent with a move away from reliance on traditional null hypothesis testing which, coupled with publication bias, may have led to inflation of published effect sizes and the "reproducibility crisis" in disciplines like Psychology [59].

The results presented in this manuscript suggest moderate intervention effects in key attention and social attention domains as well as moderate probabilities of training effects being positive (see Tables 6 and 7, and Figs 3 and 4). These short-term effects were observed even though infants took part in just three training sessions for a cumulative average duration of less than 90 min of training. Overall, these results warrant further study. There is increasing evidence that at-risk populations like VP infants are responsive to cognitive training [29], and that this type of training may be particularly effective when delivered at early ages [31], indicating cognitive control training as a promising avenue of intervening in at-risk or vulnerable populations. Furthermore, while VP infants may be at risk of learning and cognitive problems due to early birth and vulnerability to brain connectivity anomalies [60, 61], biological models suggest that early experiences may modulate individuals' reactivity to the environment [62]: VP infants' may be more susceptible to environmental inputs and thus display increased gains

from positive environmental influences [63]. Future trials will be pivotal in further investigating the effects of early cognitive interventions, as well as the durability and scope of these effects across larger and more diverse samples of VP infants.

## Supporting information

**S1 Checklist. CONSORT 2010 checklist of information to include when reporting a pilot or feasibility trial.**
(DOC)

**S1 File. Supplementary material ACT results.**
(PDF)

**S2 File. A feasibility study of the Attention Control Training (ACT) intervention amongst very preterm (VP) infants.**
(DOCX)

**S3 File. Consent form for publication in a PLOS journal.**
(PDF)

## Acknowledgments

We are very grateful to all the parents and their children who gave their time to take part in this study. Dr Nita Saxeena (HSC South-East Trust) and Dr Sanjeev Bali (HSC North Trust) gave a significant and generous contribution to the recruitment of participants in the study. Dr Nicola Doherty (HSC Western Trust) and Danielle Barnes contributed with insights and feedback during Steering Group meetings. We also wish to thank the TinyLife workers that enthusiastically helped with recruitment and the running of the study. We thank Aaron Patterson for his contribution to data collection, and Dr Ahmet Butun for his contribution to data coding.

### Sponsor of the study

Governance, Ethics and Integrity Queen's University Belfast, University Road, Belfast, BT7 1NN, UK.

## Author Contributions

**Conceptualization:** Oliver Perra, Fiona Alderdice, David Sweet, Sam Wass.

**Data curation:** Oliver Perra, Matthew Johnston.

**Formal analysis:** Oliver Perra, Sam Wass.

**Funding acquisition:** Oliver Perra, Fiona Alderdice, David Sweet, Alison McNulty, Kostas Papageorgiou, Sam Wass.

**Investigation:** Oliver Perra, Matthew Johnston, Delfina Bilello, Sam Wass.

**Methodology:** Oliver Perra, Fiona Alderdice, David Sweet, Sam Wass.

**Project administration:** Oliver Perra, David Sweet, Matthew Johnston, Delfina Bilello.

**Resources:** Oliver Perra, Alison McNulty, Matthew Johnston, Delfina Bilello.

**Software:** Sam Wass.

**Supervision:** Oliver Perra.

**Validation:** Oliver Perra.

**Visualization:** Oliver Perra.

**Writing – original draft:** Oliver Perra.

**Writing – review & editing:** Oliver Perra, Fiona Alderdice, David Sweet, Alison McNulty, Matthew Johnston, Delfina Bilello, Kostas Papageorgiou, Sam Wass.

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
