## [Decision Letter · Decision Letter 0]

7 Feb 2022

PONE-D-21-39854Attention and Social Communication Skills of Very Preterm Infants after Training Attention Control:  Findings from a Feasibility StudyPLOS ONE

Dear Dr. Perra,

Thank you for submitting your manuscript to PLOS ONE. After careful consideration, we feel that it has merit but does not fully meet PLOS ONE’s publication criteria as it currently stands. Therefore, we invite you to submit a revised version of the manuscript that addresses the points raised during the review process.

We look forward to receiving your revised manuscript.

Kind regards,

Walid Kamal Abdelbasset, Ph.D.

Academic Editor

PLOS ONE

Journal Requirements:

[We are very grateful to all the parents and their children who gave their time to take part in this study. Dr Nita Saxeena (HSC South-East Trust) and Dr Sanjeev Bali (HSC North Trust) gave a significant and generous contribution to the recruitment of participants in the study. Dr Nicola Doherty (HSC Western Trust) and Danielle Barnes contributed with insights and feedback during Steering Group meetings. We also wish to thank the TinyLife workers that enthusiastically helped with recruitment and the running of the study.  We thank the Health and Social Care Research and Development Division, Public Health Agency, for the financial support received. We are particularly grateful to Dr Julie McCarroll (PHA) for her advice and guidance. We thank Aaron Patterson for his contribution to data collection, and Dr Ahmet Butun for his contribution to data coding.]

[Financial Disclosure Statement: this study was funded by a public health agency (PHA), health and social care (HSC)  research and development division (https://research.hscni.net/) enabling research award to Dr O Perra (principal investigator) and co-investigators Prof F Alderdice, Dr S Wass, Dr K Papageorgiou, Mrs A Mcnulty, reference: STL/5274/16. 

The funders had no role in study design, data collection and analysis, decision to publish, or preparation of the manuscript.]

5. We note that Section 3 of Supplementary Material includes an image of a participant in the study. 

Reviewers' comments:

Reviewer's Responses to Questions

**Comments to the Author**

1. Is the manuscript technically sound, and do the data support the conclusions?

Reviewer #1: Yes

Reviewer #2: Yes

Reviewer #3: Yes

2. Has the statistical analysis been performed appropriately and rigorously? 

Reviewer #1: Yes

Reviewer #2: Yes

Reviewer #3: I Don't Know

3. Have the authors made all data underlying the findings in their manuscript fully available?

Reviewer #1: Yes

Reviewer #2: Yes

Reviewer #3: Yes

4. Is the manuscript presented in an intelligible fashion and written in standard English?

Reviewer #1: Yes

Reviewer #2: Yes

Reviewer #3: Yes

5. Review Comments to the Author

Reviewer #1: Review comments on Manuscript Number: (PONE-D-21-39854) entitled '' Attention and Social Communication Skills of Very Preterm Infants after Training Attention Control: Findings from a Feasibility Study''.

I believe this is potentially an interesting and useful research. Overall, this study provides a novel approach. However, the following concerns will need to be addressed.

Title:

Well structured

Abstract

- It is highly recommended to write the abstract in more structured way (background, methods, results, conclusion)

- It is recommended to write keywords (5 to 7 keywords).

Background (the Introduction)

- Well structured.

Methods

- Satisfactory

Statistical analysis

- Satisfactory

Discussion

- Satisfactory

- The limitations should be clarified in a separate section at the end of discussion.

References

- Some references need to be updated

Reviewer #2: The paper submitted by the authors is interesting and well-written. The statistical analysis is well-done and the results are illustrated in a good way. Therefore, the paper can be accepted in its current form.

Reviewer #3: The manuscripts looks to be good but have some feedbacks for minor correction:

1. in short title: Better not use the abbreviation VP.

2. Usually keywords are <6.

3. Bayesian analysis is more explained in introduction which need to be in method section.

4. Sample size is very small to generalize the study findings.

5. Will you clarify the intervention received by control group?

6. It would be good to start the 1st paragraph of the discussion with aim and hypothesis .

7. In line 302 to 307, 322 it would be compare and contras the findings with other study.

8. Though the participants were of corrected age of <12 months, what other criteria were there for inclusion? like delay developmental milestones or normal hand function like gripping .....

6. PLOS authors have the option to publish the peer review history of their article (what does this mean?). If published, this will include your full peer review and any attached files.

Reviewer #1: No

Reviewer #2: No

Reviewer #3: No

---

## [Author Response · Author response to Decision Letter 0]

25 Apr 2022

Reviewer #1: I believe this is potentially an interesting and useful research. Overall, this study provides a novel approach. However, the following concerns will need to be addressed. Title: Well structured Abstract - It is highly recommended to write the abstract in more structured way (background, methods, results, conclusion)

Answer: We thank the reviewer for the positive feedback and the comments. We have revised the Abstract (Manuscript Page 2) as suggested.

- It is recommended to write keywords (5 to 7 keywords).

Answer: We have provided a list of keywords after the Abstract (see Manuscript, Page 3).

Background (the Introduction) - Well structured. Methods - Satisfactory Statistical analysis - Satisfactory Discussion - Satisfactory - The limitations should be clarified in a separate section at the end of discussion.

Answer: We have clearly identified the section on limitations providing a sub-title and moving this at the end of the Discussion, as suggested (Manuscript, Page 26).

References - Some references need to be updated

Answer: We updated these. In addition, we have provided references to recent key papers such as: Kaul et al. 2021, DOI: 10.1038/s41390-021-01895-8; Wheelock et al. 2021, DOI: 10.1093/cercor/bhaa303; Eves et al., 2021, DOI: 10.1097/DBP.0000000000000806.

Reviewer #2 Reviewer #2: The paper submitted by the authors is interesting and well-written. The statistical analysis is well-done and the results are illustrated in a good way. Therefore, the paper can be accepted in its current form.

Answer: We thank the reviewer for the praise received and their time in reviewing our manuscript.

Reviewer #3 The manuscripts looks to be good but have some feedbacks for minor correction: 1. in short title: Better not use the abbreviation VP.

Answer: We thank the reviewer for the positive feedback. We have accepted this suggestion and corrected the short title. 2. Usually keywords are <6.

Answer: We have provided a list of keywords after the Abstract (Manuscript, Page 3). 3. Bayesian analysis is more explained in introduction which need to be in method section.

Answer: We have thought carefully about this comment. We have however come to the conclusion that a presentation of the key principle underlying Bayesian approaches and how these differ from more traditional approaches is necessary because: (a) The manuscript aims to make a case for the former approaches being used more widely, particularly in exploratory and small-sample analyses; (b) Bayesian approaches are still not widely recognised across disciplines. We thus believe that a relatively short presentation (2 paragraphs) of Bayesian approaches in the Introduction contributes to clarity and dissemination of the paper among a wider audience. 4. Sample size is very small to generalize the study findings.

Answer: We have further emphasised and highlighted this point in the “Limitations” section in the Discussion (Page 26) 5. Will you clarify the intervention received by control group?

Answer: We have restructured the “Interventions” section in the Methods (Page 9 and 10) to increase clarity and provide more details about the control procedure. 6. It would be good to start the 1st paragraph of the discussion with aim and hypothesis .

Answer: Thank you for this comment, which we have taken into account, revising the Discussion accordingly (Page 23). 7. In line 302 to 307, 322 it would be compare and contras the findings with other study.

Answer: We have provided some comparisons to two studies that have used similar procedures in the delivery of the intervention as well as similar measures, in Page 23, as suggested. 8. Though the participants were of corrected age of <12 months, what other criteria were there for inclusion? like delay developmental milestones or normal hand function like gripping .....

Answer: We have highlighted these criteria more clearly in the Methods section (Page 6). In the Limitations section of the Discussion (Page 26) we have also highlighted that the inclusion/exclusion criteria determined the sample was made up of healthy VP infants.

---

## [Decision Letter · Decision Letter 1]

9 May 2022

PONE-D-21-39854R1Attention and Social Communication Skills of Very Preterm Infants after Training Attention Control:  Findings from a Feasibility StudyPLOS ONE

Dear Dr. Perra,

Thank you for submitting your manuscript to PLOS ONE. After careful consideration, we feel that it has merit but does not fully meet PLOS ONE’s publication criteria as it currently stands. Therefore, we invite you to submit a revised version of the manuscript that addresses the points raised during the review process.

We look forward to receiving your revised manuscript.

Kind regards,

Walid Kamal Abdelbasset, Ph.D.

Academic Editor

PLOS ONE

Journal Requirements:

Reviewers' comments:

Reviewer's Responses to Questions

**Comments to the Author**

1. If the authors have adequately addressed your comments raised in a previous round of review and you feel that this manuscript is now acceptable for publication, you may indicate that here to bypass the “Comments to the Author” section, enter your conflict of interest statement in the “Confidential to Editor” section, and submit your "Accept" recommendation.

Reviewer #1: All comments have been addressed

Reviewer #3: All comments have been addressed

2. Is the manuscript technically sound, and do the data support the conclusions?

Reviewer #1: Yes

Reviewer #3: Yes

3. Has the statistical analysis been performed appropriately and rigorously? 

Reviewer #1: Yes

Reviewer #3: Yes

4. Have the authors made all data underlying the findings in their manuscript fully available?

Reviewer #1: Yes

Reviewer #3: Yes

5. Is the manuscript presented in an intelligible fashion and written in standard English?

Reviewer #1: Yes

Reviewer #3: Yes

6. Review Comments to the Author

Reviewer #1: Review comments on Manuscript Number: PONE-D-21-39854R1. Entitled "Attention and Social Communication Skills of Very Preterm Infants after Training Attention Control: Findings from a Feasibility Study"

I would like to thank the authors for their successful work to address the reviewers' comments. The authors have done great efforts to accomplish this work. They fulfilled all comments and made necessary changes throughput the manuscript. I recommend accepting the manuscript its revised form.

Reviewer #3: Now the manuscript looks to be good and I have just a few minor corrections:

1. In referencing, Plosone journal uses the bracket [1] for citation. Just see the articles published in this journal.

2. Line 58 to 68 describes the procedures in the introduction part which need to be in the method section.

3. In the consort flow diagram, check the size of boxes as some words are hidden.

7. PLOS authors have the option to publish the peer review history of their article (what does this mean?). If published, this will include your full peer review and any attached files.

Reviewer #1: No

Reviewer #3: **Yes: **Bishnu Dutta Acharya

---

## [Author Response · Author response to Decision Letter 1]

16 Jun 2022

Reviewer #3: 

Now the manuscript looks to be good and I have just a few minor corrections:

1. In referencing, Plosone journal uses the bracket [1] for citation. Just see the articles published in this journal.

Answer: Thank you for drawing our attention to this. We have carried out this change throughout the manuscript. 

2. Line 58 to 68 describes the procedures in the introduction part which need to be in the method section.

Answer: We have now moved this section to the start of the Methods. 

3. In the consort flow diagram, check the size of boxes as some words are hidden.

Answer: We apologise for this mistake, which we have now corrected.

---

## [Decision Letter · Decision Letter 2]

24 Jun 2022

PONE-D-21-39854R2Attention and Social Communication Skills of Very Preterm Infants after Training Attention Control:  Findings from a Feasibility StudyPLOS ONE

Dear Dr. Perra,

Thank you for submitting your manuscript to PLOS ONE. After careful consideration, we feel that it has merit but does not fully meet PLOS ONE’s publication criteria as it currently stands. Therefore, we invite you to submit a revised version of the manuscript that addresses the points raised during the review process.

We look forward to receiving your revised manuscript.

Kind regards,

Walid Kamal Abdelbasset, Ph.D.

Academic Editor

PLOS ONE

Journal Requirements:

Reviewers' comments:

Reviewer's Responses to Questions

**Comments to the Author**

1. If the authors have adequately addressed your comments raised in a previous round of review and you feel that this manuscript is now acceptable for publication, you may indicate that here to bypass the “Comments to the Author” section, enter your conflict of interest statement in the “Confidential to Editor” section, and submit your "Accept" recommendation.

Reviewer #3: All comments have been addressed

2. Is the manuscript technically sound, and do the data support the conclusions?

Reviewer #3: Yes

3. Has the statistical analysis been performed appropriately and rigorously? 

Reviewer #3: Yes

4. Have the authors made all data underlying the findings in their manuscript fully available?

Reviewer #3: Yes

5. Is the manuscript presented in an intelligible fashion and written in standard English?

Reviewer #3: Yes

6. Review Comments to the Author

Reviewer #3: Author have addressed the reviewer's comments.

This paper looks already published with the same protocol number and similar title although the statical findings are different.

Please confirm it once. The published paper title is "Very preterm infants engage in an intervention to train their control of attention: results from the feasibility study of the Attention Control Training (ACT) randomised trial".

PMC7952829

The published article is attached with this.

7. PLOS authors have the option to publish the peer review history of their article (what does this mean?). If published, this will include your full peer review and any attached files.

Reviewer #3: **Yes: **Bishnu Dutta Acharya

---

## [Author Response · Author response to Decision Letter 2]

3 Aug 2022

Reviewer #3 raised concerns regarding the originality of the study, highlighting results from the same study have been published in a previous paper titled “Very preterm infants engage in an intervention to train their control of attention: Results from the feasibility study of the Attention Control Training (ACT) randomised trial”, published in Pilot and Feasibility Studies, Volume 7, 23, p. 66.

We thank the reviewer for these comments, and take the chance to highlight that while the results are from the same study, the previous publication referred to the outcomes of the study that concerned its feasibility, i.e.: recruitment and retention, engagement with the training, acceptability and completion of outcome measures. Conversely, in the manuscript we have submitted to PLOS One, we have focused on the infants’ performance in the outcome measures collected and used a relatively novel approach, Bayesian regression, to estimate the direction and size of training effects. 

We appreciate the reviewer’s comments, and to ensure that we emphasise the different content and scope of the two papers, we have carried out the following changes to the manuscript:

1. We have changed the title from “Attention and Social Communication Skills of Very Preterm Infants after Training Attention Control: Findings from a Feasibility Study” to “Attention And Social Communication Skills of Very Preterm Infants after Training Attention Control: Bayesian Analyses of a Feasibility Study”.

2. We have added a sentence in the Abstract (Page 2, Line 37), which now reads:

In a previous study we analysed the data concerning feasibility (e.g. recruitment and retention). In the paper presented here we considered the infants’ performance and used Bayesian regression in order to provide credible treatment estimates considering the data collected. 

3. We have added a sentence in the Introduction (Page 5, Line 5), which now reads: 

We conducted a study to test the feasibility of delivering the ACT to VP infants [36]: in a previous study [37,38] we presented the main results concerning feasibility (e.g. recruitment and retention): these results indicated that infants engaged in the training and assessments. Conversely, in this manuscript we investigate the direction of treatment effects across a series of infants’ outcomes using Bayesian regression. Bayesian analyses allow to update assumptions about parameters (e.g. differences between two groups) by incorporating new information [39].

4. We have added a sentence in the Discussion (Page 24, Line 12), which now reads:

This is the first study that trained VP infants’ attention control. Previous analyses indicated it is possible to recruit and retain VP infants and engage them in the training [37,38]. Conversely, the analyses of infants’ performance in different tasks presented in this paper are important in providing initial estimates concerning the potential for improving early key cognitive abilities of this at-risk group.

All these changes are meant to highlight that a previous paper has been published that reports the results concerning feasibility of the study and, conversely, the current manuscript concerns the use of Bayesian methods to explore the direction of treatment effects on infants’ abilities, which were not considered as outcomes in the previously published paper.

We hope these changes address the concerns raised by the reviewer, increasing clarity for the readership. We thank the reviewer and the Editor again for the chance to improve our manuscript, and look forward to hearing from you.

Sincerely

Dr Oliver Perra

---

## [Decision Letter · Decision Letter 3]

16 Aug 2022

Attention and Social Communication Skills of Very Preterm Infants after Training Attention Control:  Bayesian analyses of a Feasibility Study

PONE-D-21-39854R3

Dear Dr. Perra,

We’re pleased to inform you that your manuscript has been judged scientifically suitable for publication and will be formally accepted for publication once it meets all outstanding technical requirements.

Kind regards,

Walid Kamal Abdelbasset, Ph.D.

Academic Editor

PLOS ONE

Additional Editor Comments (optional):

Reviewers' comments:

Reviewer's Responses to Questions

**Comments to the Author**

1. If the authors have adequately addressed your comments raised in a previous round of review and you feel that this manuscript is now acceptable for publication, you may indicate that here to bypass the “Comments to the Author” section, enter your conflict of interest statement in the “Confidential to Editor” section, and submit your "Accept" recommendation.

Reviewer #3: All comments have been addressed

2. Is the manuscript technically sound, and do the data support the conclusions?

Reviewer #3: Yes

3. Has the statistical analysis been performed appropriately and rigorously? 

Reviewer #3: Yes

4. Have the authors made all data underlying the findings in their manuscript fully available?

Reviewer #3: Yes

5. Is the manuscript presented in an intelligible fashion and written in standard English?

Reviewer #3: Yes

6. Review Comments to the Author

Reviewer #3: Thank you for addressing the previous comments. The article looks fine and there looks difference from previous publications.

7. PLOS authors have the option to publish the peer review history of their article (what does this mean?). If published, this will include your full peer review and any attached files.

Reviewer #3: **Yes: **Bishnu Dutta Acharya

---

## [Editor Report · Acceptance letter]

12 Sep 2022

PONE-D-21-39854R3 

Attention and social communication skills of very preterm infants after training attention control: Bayesian analyses of a feasibility study 

Dear Dr. Perra:

I'm pleased to inform you that your manuscript has been deemed suitable for publication in PLOS ONE. Congratulations! Your manuscript is now with our production department. 

Kind regards, 

on behalf of

Dr. Walid Kamal Abdelbasset 

Academic Editor

PLOS ONE